# Exploring the Potential Role of Ribosomal Proteins to Enhance Potato Resilience in the Face of Changing Climatic Conditions

**DOI:** 10.3390/genes14071463

**Published:** 2023-07-18

**Authors:** Eliana Valencia-Lozano, Lisset Herrera-Isidrón, Jorge Abraham Flores-López, Osiel Salvador Recoder-Meléndez, Braulio Uribe-López, Aarón Barraza, José Luis Cabrera-Ponce

**Affiliations:** 1Departamento de Ingeniería Genética, Centro de Investigación y de Estudios Avanzados del IPN, Unidad Irapuato, Irapuato 36824, Guanajuato, Mexico; eliana.valencia@cinvestav.mx; 2Unidad Profesional Interdisciplinaria de Ingeniería Campus Guanajuato (UPIIG), Instituto Politécnico Nacional, Av. Mineral de Valenciana 200, Puerto Interior, Silao de la Victoria 36275, Guanajuato, Mexico; lherrerai@ipn.mx (L.H.-I.); jfloresl1701@alumno.ipn.mx (J.A.F.-L.); orecoderm1601@alumno.ipn.mx (O.S.R.-M.); buribel1900@alumno.ipn.mx (B.U.-L.); 3CONACYT-Centro de Investigaciones Biológicas del Noreste, SC., Instituto Politécnico Nacional 195, Playa Palo de Santa Rita Sur, La Paz CP 23096, Baja California Sur, Mexico; abarraza@cibnor.mx

**Keywords:** microtubers, potato, ribosomal proteins, transcriptome analysis

## Abstract

Potatoes have emerged as a key non-grain crop for food security worldwide. However, the looming threat of climate change poses significant risks to this vital food source, particularly through the projected reduction in crop yields under warmer temperatures. To mitigate potential crises, the development of potato varieties through genome editing holds great promise. In this study, we performed a comprehensive transcriptomic analysis to investigate microtuber development and identified several differentially expressed genes, with a particular focus on ribosomal proteins—RPL11, RPL29, RPL40 and RPL17. Our results reveal, by protein–protein interaction (PPI) network analyses, performed with the highest confidence in the STRING database platform (v11.5), the critical involvement of these ribosomal proteins in microtuber development, and highlighted their interaction with PEBP family members as potential microtuber activators. The elucidation of the molecular biological mechanisms governing ribosomal proteins will help improve the resilience of potato crops in the face of today’s changing climatic conditions.

## 1. Introduction

Potato (*Solanum tuberosum* L.) is a staple food that produces tubers with high nutritional value and energy. It is the fourth most important crop and its yield will be affected by extreme drought, heat, and the relocation of potato diseases and pests. Food security depends on the development of strategies to overcome the effects of climate variability on the agriculture system [1]. Potatoes are propagated vegetatively from harvested tubers and certified seeds from suppliers. An understanding of the tuberization process is needed to boost resilience in the face of changing climatic conditions.

The molecular mechanisms involved in potato tuber development under greenhouse conditions have been widely studied for review; see [2]. An alternative technology is the in vitro induction of potato tubers called microtubers (MTs). MTs induction occurs when axillary buds are cultured in a medium containing high sucrose content, plant growth regulators, and different light quality/darkness. MTs production has advantages in terms of storage, transport, and cultivation due to their reduced size and weight, requiring less space than seedlings, with higher multiplication rates producing seed potatoes faster and cheaper than other methods [3,4,5].

In our previous work, we developed a protocol for MTs induction in potato *S. tuberosum* var α using stolons cultured in MS medium supplemented with 8% sucrose, gelrite 6.0 g/L, 2iP 10 mg/L under darkness conditions. The rationale for this protocol was that CK signaling interacts with homeobox transcription factors, RPs, cell cycle, carbon metabolism, auxin-responsive factors and stem cell genes in the microtuberization process [6].

In our following work, we reported a transcriptome analysis of the MTs revealing that 1715 up-regulated and 1624 down-regulated genes were involved in this biological process. The protein–protein interaction (PPI) network analyses, performed with the highest confidence in the STRING database platform (v11.5, www.string-db.org accessed on 15 September 2022), showed that 299 genes were tightly associated in 14 clusters including a main group of essential life proteins.

Ribosomal proteins (RPL11, RPL29, RPL40, RPL17) interact with several gene clusters, like the tuberigen, proteasome, immunophilins and oxidative stress. RPL11 interacts with thioredoxins, thylakoids, fatty acid biosynthesis, one-carbon metabolism, carbon metabolism, flowering (six PEBP family members), and the cell cycle [7].

Through a yeast two-hybrid screening approach, the interaction of PEBP-StSP6A (a positive regulator of tuber development) [8,9,10] with RPs and others involved in protein synthesis, RNA and DNA binding proteins, histones, initiation factors, signaling and carbon metabolism has been demonstrated [11,12].

Several transcriptomic analyses of potato tuberization have revealed the presence of RPs during the process [13,14,15,16,17]. From the RNA-seq database published by Sharma and Hannapel 2016 [13], we performed a PPI network analysis with the highest confidence (0.900), revealing that RPs interact with SP6A, RP40SA, RP60S, RPS8, RPS4A, RPL10, BEL5, and RPL14 [13].

In this manuscript, we will discuss the feasibility of activating RPs to induce MTs with several trait advantages such as size, number, stress tolerance, and protein content. We will also discuss the function and potential use of several genes interacting with RPs derived from the transcriptome analysis performed by Valencia-Lozano et al. in 2022 [7].

## 2. Materials and Methods

### 2.1. Potato MTs Induction

Potato MTs induction was induced by culture of stolon explants, about 3 cm in length, in flasks containing medium: MS medium, supplemented with 2iP 10 mg/L, 8% sucrose, 6 g/L gelrite, activated charcoal (0.3%), and pH 5.8 identified as MR8-G6-2iP. In the control medium, MS medium was supplemented with 1% sucrose, 3 g/L gelrite and 10 mg/L 2iP identified as MR1-G3-2iP. Flasks were sealed with plastic and incubated in the dark at 25/17 °C for 15 days [6].

### 2.2. RNA Isolation and qPCR Analysis

Total RNA derived from 15-day-old explants was isolated according to the methods of Valencia-Lozano et al. (2021) [7]. To validate the genome-wide analysis of MTs, oligonucleotides were designed (Table 1), and the expressions of *EF1* and *SEC3* were used as reference genes for calculating the relative amount of target gene expression using the 2^−ΔΔCT^ method [18,19]. qPCR analysis was based on at least five biological replicates for each sample with three technical replicates.

The sequencing of cDNA derived from potato MTs and controls was undertaken in GENEWIZ, Plainfield, NJ, USA. To sequence, the Illumina HiSeq 4000 (Illumina, San Diego, CA, USA) was applied. The quality of sequence reads was assessed by the software package FastQC (http://www.bioinformatics.babraham.ac.uk/projects/fastqc/ accessed on 2 June 2023) and to remove sequence adapters and low-quality bases we used the software Trimmomatics [20].

### 2.3. Transcriptome and Interaction Analysis of Proteins Involved in Microuberization

RNA-seq reads aligned to the potato *S. tuberosum* reference genome were made according to the methods set out by Valencia-Lozano et al. (2022) [7]. The quantification and differential analysis of the transcripts was performed using the DESeq2 v1.12.4 program. Finally, an ontology analysis was performed using Blast2GO. The PPI analysis of microtuberization was performed with the STRING database v11.5 [21], based on *S. tuberosum* and the highest confidence (0.860).

## 3. Results

### 3.1. Transcriptome Analysis of MTs Induction

The transcriptome analysis of the MTs induction process was analyzed according to Valencia-Lozano et al. (2022) [7]. The study revealed 1699 up-regulated genes, and when analyzed in a PPI network with the highest confidence, 299 were tightly associated with two fundamental biological processes essential for life and highly conserved through organisms: RPs comprising 29 proteins and CC containing 117 proteins (Figure 1).

RPs interact with proteins that sense the environment: six PEBP family members, twenty-one involved in osmotic stress, nine related to oxidative stress, twenty three involved in CK response, six related to one-carbon metabolism, thirty eight associated with carbon metabolism, sixteen with TCA cycle, six with acyl carrier proteins, fourteen involved in fatty acid metabolism, thirteen in thylakoid, nine redoxins, eight involved in sulfur metabolism, five involved in disulfide isomerase activity and twelve related to immunophilins (Figure 1).

### 3.2. DEG in MTs Development Involved in Immunophilins, Redoxins, Oxidative Stress, Carbon Metabolism and One-Carbon Metabolism

In the PPI network, 11 genes were involved in immunophilins (Figure 2), 8 in redoxins (Figure 3), 8 in oxidative stress (Figure 4), 38 in carbon metabolism (Figure 5) and 6 in one-carbon metabolism (Figure 6). This cluster interacts with the cell cycle cluster through the interaction of the dihydrofolate reductase gene and RPs.

### 3.3. Validation of the Transcriptome-Wide Analysis

Validation in the transcriptomic-wide analysis was achieved by selecting the following genes: *StSP6A* (PGSC0003DMT400060057), *RPL11* (PGSC0003DMT400031869), *RPL29* (PGSC0003DMT400069470), *RPL40* (PGSC0003DMT400047686) and *RPL17 (PGSC0003DMT400060127*). For DEG validation, *EFa1* and *SEC3* were used as endogen controls. The results of the validation indicate that the values are consistent with those obtained in the transcriptomic-wide analysis (Figure 7).

## 4. Discussion

### 4.1. What Is the Importance of Ribosome Biogenesis?

Ribosome biogenesis is tightly associated with plant growth, development, and reproduction. Genetic mutations related to ribosomal proteins (RPs) or ribosome biogenesis factors (RBFs) result in retarded growth, delayed flowering, and in more severe cases, they are lethal. In total, 19 ribosomal proteins resulted in loss of function and 26 ribosome biogenesis factors are seedling/embryo-lethal (Table 2).

#### Some RPs Are Lethal Mutants

The small-subunit RPs exhibit lethal effects; for example, some are seedling-lethal, such as RPS1 and RPS20 in rice [22,29], and RPS18A in tobacco [28], and some are embryo-lethal, such as RPS9 and RPS17 in maize [24,27], and RPS16 and RPS27 in Arabidopsis [26,31] (Table 2).

Large-subunits RPs, such as RPL13, RPL12, RPL13 and RPL21C, are seedling-/embryo-lethal in rice [36,37,77,78] (Table 2). In contrast, RPL5C, RPL9C,D, RPL10, RPL21C, and RPL28-1 are embryo-lethal in Arabidopsis [32,33,34,35,79] (Table 2). It has been shown that at least 29 ribosome biogenesis are lethal, including proteins involved in chloroplast development, ribosome biogenesis, nucleolar organization, and chlorophyll biosynthesis (Table 2).

In our transcriptome analysis, five RPLs (RPL1, 12, 13, 27 and 35), and four RPSs (RPS1, 9, 16 and 17) are embryo-lethal.

### 4.2. Overexpression of Ribosomal Proteins and Ribosome Biogenesis Factors

Horvath et al. (2006) [80] demonstrated the highest expression of *EBP1* in the developing organs and its correlation with genes involved in ribosome biogenesis in potatoes. The *EBP1* gene regulates the intermediate and late steps of rRNA processing. Silenced potato lines showed reduced size and tuber yield, and an abnormal morphology.

Transformed potato plants with the RP *StoL13a* from *Solanum torvum*, SW, a highly resistant plant to *Verticillium dahliae* infection, were more resistant to *V. dahliae* infection than the control plants. The transgenic plants showed lower levels of reactive oxygen species and attenuated oxidative damage. In addition, six defense and antioxidant enzyme genes were up-regulated in the *StoL13a* ectopic expression plants. These results suggest that StoL13a plays a role in plant defense against *V. dahliae* infection [81].

The overexpression of RPL6 in rice resulted in salt tolerance [82]. Transgenic rice plants overexpressing RPL23A showed resistance to water use efficiency, and suitable growth and yield parameters, compared to the negative control [83].

*OLI2/NOP2A* encodes a nucleolar methyltransferase required to mature the 25S ribosomal RNA of the 60S large ribosomal subunit. These seeds were lighter and heavier than wild-type seeds produced by *oli2* mutant and *OLI2* overexpressor plants respectively. The seeds from the *oli2* mutant showed delayed germination, while *OLI2* overexpressor lines germinated earlier than the wild type. The wild type had a greater number and longer length of lateral roots than the *oli2* mutant. The lateral root development phenotype of the *oli2* mutant resembles auxin-related mutants, but was not enhanced by exogenously supplied auxin. Furthermore, high concentrations of sugar induced hypersensitivity and less sensitivity in oli2 mutant and *OLI2* overexpressor lines, respectively [84].

*STCH4/REIL2* encodes a ribosomal biogenesis factor up-regulated during cold stress. When *STCH4* is overexpressed, it confers chilling and freezing tolerance to Arabidopsis, although its mutation reduces CBF protein levels, resulting in the delayed induction of C-repeat-binding factor (CBF) regulon genes [85].

### 4.3. Are RPs Good Candidate Genes for Improving of Multiple Abiotic Stress Tolerance in Potato?

Drought stress affects potato plants at all stages of the crop’s growth, from seedling emergence to tuber initiation, and bulking, ultimately resulting in a reduction in tuber yield. Prolonged water scarcity induces several physiological disorders in potato tubers, such as tuber cracking, tuber malformation, hollow heart, vascular discoloration, and reduction in dry matter accumulation.

Kappachery et al. (2013) [86] identified potential drought tolerance genes in potato using a yeast functional screening method. A cDNA expression library was constructed from hyperosmotic stressed potato plants, and identified yeast transformants expressing different cDNAs for survival under hyperosmotic stress conditions.

Sixty-nine genes were identified to grow under drought, salt, and heat stress. Of these, eight were RPs (RPS7, RPL12, RPL10, RPL27, RPS11, RPL18a, RPL1 and RPL36) [86]. The potential of RPs to obtain potato MTs lines with desirable traits is very promising.

### 4.4. Interaction of RPs Cluster with Immunophilins

In the PPI network, 11 genes affected immunophilins. RPL29 (PGSC0003DMT400069847) interacts with peptidyl-prolyl cis-trans isomerase *FKBP12*. This group of proteins includes peptidyl-prolyl cis-trans-isomerases (PPIs) (immunophilins) and protein disulfide isomerases (Figure 8).

The cluster consists of four PPI genes interacting with *SOS3* (PGSC0003DMT400023568), a calcium sensor calcineurin B essential for the transduction of the salt stress-induced Ca_2+_ signal and salt tolerance in Arabidopsis. A loss-of-function mutation that reduces the Ca_2+_ binding capacity of *SOS3* (*sos3-1*) renders the mutant hypersensitive to salt [87].

*FKBP12*, overexpressed in Arabidopsis, has been reported to be directly involved in abiotic stress responses and in promoting growth under normal conditions [88].

The overexpression in potatoes of cyclophilin *CYP21-4* involved in oxidative stress formed longer plants and heavier tubers, and when microtubers were induced, they yielded more in a shorter time [89].

Three genes with disulfide isomerase-like proteins are present in this cluster—*STPDI1*, 2 and 3. They interact with calnexin (PGSC0003DMT400036920). *StPDI1*, accumulated upon salt exposure, catalyzes the formation of disulfide bonds and confers tertiary and quaternary structures to the proteins. Thus, they may act as chaperones during salt stress, as previously described for HSPs [90].

The silencing of *StPDI1* expression affects the tolerance of abiotic stress in transgenic potato plants. The amount of malate, succinate and 2-oxoglutarate, and the content of the reducing equivalent NADH, were decreased significantly in *StPDI1*-inhibited potato plants. In contrast, amino acids such as serine and threonine were upregulated compared to wild-type plants [91].

### 4.5. Redoxins

To minimize the adverse effects of reactive oxygen species (ROS), aerobic organisms have evolved defense systems, catalases (*CAT*), superoxide dismutase (*SOD*), ascorbate peroxidase (*APX*), glutathione peroxidase (*GPX*) and *GST,* as well as the capacity for the production of low-molecular weight antioxidants such as ascorbic acid (AA) and glutathione (GSH).

In the PPI network, 9 genes affected redoxins. RPs interact with the thioredoxin cluster through the interaction of the *PRXQ* peroxiredoxin (PGSC0003DMT400035271) with RPL11 (PGSC0003DMT400031869) and PRL18 (PGSC0003DMT400011931). *PRXQ* plays a role in protecting cells from oxidative stress by detoxifying peroxides. Also, this protein is involved in photosystem II’s protection against hydrogen peroxide.

Transgenic potato plants expressing 2-cysteine peroxiredoxin exhibit enhanced tolerance to environmental stresses, including MV-induced oxidative stress and high temperature, with plants under the control of the SWPA2 promoter exhibiting the best tolerance [92].

### 4.6. Response to Oxidative Stress

Within a cell, the superoxide dismutases (SODs) form the primary defense against ROS. In the PPI network, nine genes were involved in oxidative stress. Reactive O_2_ species (ROS) are produced in cells under both unstressed and stressed conditions.

The superoxidase dismutase 1 gene is essential to the potato’s response to low temperatures. The overexpression of *StSOD1* increases low-temperature tolerance in potatoes, while interference expression decreases low-temperature tolerance in potatoes [93]. The overexpression of a cytosolic copper-zinc superoxide dismutase, from *Potentilla atrosanguinea* (*PaSOD*), in potato (*S. tuberosum* ssp. *tuberosum* L. cv. Kufri Sutlej) enhanced net photosynthetic rates (PN) and stomatal conductance (gs) compared to those in the wild-type (WT) plants under control (irrigated) as well as drought stress conditions [94]. Transgenic potato plants overexpressing *SOD* and ascorbate peroxidase (*APX*) promoted the enhancement of lignification and starch biosynthesis, and improved salt tolerance [95]. The overexpression in potato plants of *SOD*, *APX* and the bacterial choline oxidase (*oda*) led to enhanced protection against various abiotic stresses [96].

### 4.7. RPs Interacting with Carbon Metabolism

Carbon metabolism (CM) transforms carbon into energy at different amounts through glycolysis, gluconeogenesis, the pentose phosphate pathway, carbon fixation pathways, and the TCA pathway. In carbon metabolism, 38 genes were implicated in the PPI network. Through the interaction of the DHFR and RPs clusters, the CC cluster interacts with this cluster (Figure 9).

Watkinson et al. (2006) [97] assessed three accession genotypes of drought-stressed *S. tuberosum* ssp. *andigena* and made a transcriptomic analysis of genes associated with carbon metabolism, citrate cycle and oxidative stress. In agreement with their results, we found similar genes compared with “intermediate” genotypes, with 20 genes up-regulated under stress conditions, similar to our analysis. The term “intermediate” means plants that recovered their photosynthetic index after one cycle of stress and performed even better in the second, yielding similar results to the control plants unaffected by the stress. This may explain why molecular mechanisms under field conditions are very similar to those under in vitro conditions.

This includes *PGK*, phosphoglycerate kinase, cytosolic; *TPI*, triosephosphate isomerase, cytosolic; *IAR4*, pyruvate dehydrogenase e1 component subunit α-3, *GAPC2*, glyceraldehyde 3-phosphate dehydrogenase; *LOS2*, enolase and *PKP3*, pyruvate kinase. However, they showed that genes like Hexokinase-1 and Fructose-bisphosphate aldolase were down-regulated. In contrast, pyruvate dehydrogenase e1 component subunit β-3 did not show significant changes in expression, and these also did not occur when we induced stress. Regarding the TCA cycle, we found, according to Watkinson et al. (2006) [97], that the up-regulated genes are MMDH1 (malate dehydrogenase) and *MDH* (NAD-malate dehydrogenase). The overexpression of these genes plays a crucial role in different plant breeding programs. The phosphoglycerate kinase gene promotes biomass and yields in tobacco under salt stress conditions [98], while in rice, it improves thermotolerance [99]. Similarly, the overexpression of the pyruvate dehydrogenase gene is involved in grain size and weight in rice [100], and drought stress in barley [101] and rice [102]. Moreover, the overexpression of pyruvate kinase negatively affects root growth in maize [103], while when it is silenced in rice, this results in sucrose translocation defects, the inhibition of grain filling [104], and reduced grain starch content [105]. Additionally, the triosephosphate isomerase gene enhances photosynthesis under elevated CO_2_ levels in rice [99], makes pigeon peas more resilient to salt stress [106], and improves drought stress tolerance in both rice [107] and maize [108].

Salt stress tolerance is found to be influenced by enolase in *Mesembryanthemum crystallinum* L. Similarly, the salt stress response is tied to the fructose-bisphosphate aldose gene in *Ulva compressa* [109] and mango [110], as well as biomass accumulation in tobacco [111]. The overexpression of the acetyl-CoA carboxylase gene leads to an increased lipid content in microalgae, including *Dunaliella* sp. [112], *Chlamydomonas reinhardtii* [113], and *Scenedesmus* sp. [114], and increased seed yield in tobacco [115].

The overexpression of the sucrose phosphate synthase gene enhances growth, thermotolerance [116], sink strength in tomatoes [117], potato yield characteristics [118], and cold tolerance in chrysanthemum [119]. It also positively impacts biomass production in sugarcane [120] and in tomatoes under saturated light and CO_2_ conditions [121], and enhances foliar sucrose/starch levels in Arabidopsis [122]. Similarly, the ATP-citrate synthase gene plays a role in the salt stress response in *Halogeton glomeratus* [123] and sugar beet [124].

The overexpression of the glyceraldehyde 3-phosphate dehydrogenase gene enhances drought tolerance in potato [125] and salt tolerance in soybean [126], rice [127], and potato [128]. The 6-phosphogluconate dehydrogenase overexpression gene contributes to resistance against *Nilaparvata lugens* in rice [129], as well as salt tolerance in barley [130], and is also involved in starch accumulation in maize [131]. Additionally, malate dehydrogenase gene overexpression increases the production of organic acids and aluminum tolerance in alfalfa [132], as well as salt tolerance in rice [133], apple, and tomato, and also impacts cold tolerance [134,135]. However, the gene has been shown to be embryo-lethal in *Arabidopsis* [136].

Lastly, the overexpression of the phosphoenolpyruvate carboxylase gene increases photosynthetic efficiency in rice [137], fatty acid production in *Nicotiana tabacum* [138] and protein content in *Vicia narbonensis* [139], and affects dark and light respiration in potato [140].

In cerium stress-treated microalgae, *Nannocloropsis oculata* led to an increase in lipid content with the carbon metabolism and ribosome biogenesis genes prominently activated [141].

#### One-Carbon Metabolism

One-carbon metabolism is a critical metabolic process in which multiple enzymatic reactions provide methyl groups (one carbon) for nucleotide metabolism, purine and pyrimidine synthesis, and amino acid metabolism. These effects involve many cellular activities, such as cell growth, differentiation, and development. One-carbon metabolism transfers a carbon unit from serine or glycine to tetrahydrofolate (THF) to form methylene-THF for DNA synthesis. In the PPI network, seven genes were found to be involved in one-carbon metabolism.

The overexpression of the S-adenosylmethionine synthetase gene (*SHM4* and *SHM1*) enhances cold and salt tolerance in tobacco [142,143], lipid production in Chlamydomonas [144], salt, H_2_O_2_ and drought tolerance in Arabidopsis [145,146], and alkali tolerance in tomato [147]. The overexpression of the adenosylhomocysteinase gene (*HOG1*) results in early flowering and reduced biomass in Arabidopsis [148], and it increases lycopene and reduces ripening time in tomatoes [149]. The serine hydroxymethyltransferase gene (*MAT3*), when overexpressed in rice, confers salt tolerance [150], increases root growth and sugar levels, and decreases H_2_O_2_ levels in Arabidopsis [151], while also limiting cold tolerance [152] and antioxidant ability in rice [153].

Lastly, the bifunctional dihydrofolate reductase-thymidylate synthase (*DHFR-TS/THY-1*) can regulate folate abundance in Arabidopsis [154], which is essential for the biosynthesis of nucleotide precursors of DNA [155] and also involved somatic embryogenesis in carrot [156].

## 5. Conclusions

The gene modulation of ribosomal proteins by genome editing can increase the genetic variability to enhance potato resistance to biotic and abiotic stress, and also increase nutritional value.

The genes involved in carbon metabolism—*PGK*, *TPI*, *IAR4*, *GAPC2*, *LOS2*, and *PKP3*—in both plant field experiments and under our conditions have the potential to improve biomass and yield under stress conditions.

The cluster of immunophilins and disulfide isomerase interacting with RPs will allow for the activation of alternative mechanisms of survival enhancement under adverse conditions.

The gene modulation of one-carbon metabolism pathways will favor survival in adverse environments.

## Figures and Tables

**Figure 1 genes-14-01463-f001:**
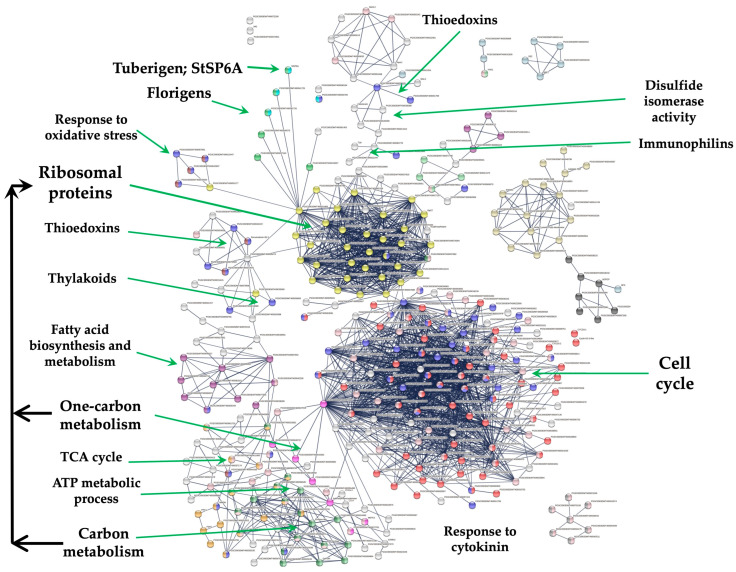
PPI network of MTs development in MR8-G6-2iP medium under darkness. In the network there is a tight interaction of the RPs cluster with the tuberigen *StSP6A* and PEBP family members, thioredoxins, immunophilins, oxidative stress, one-carbon and carbon metabolism. The PPI network has the highest confidence (0.900).

**Figure 2 genes-14-01463-f002:**
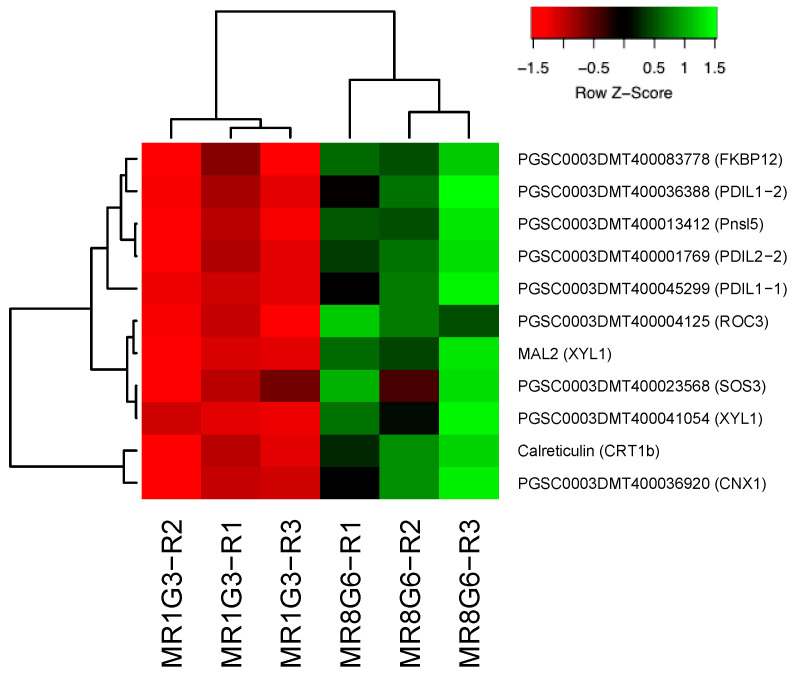
Hierarchical clustering analyses (HCA) and heat map of up-regulated genes involved in immunophilins during MTs development under dark conditions; levels of up-regulation are presented in Log2.

**Figure 3 genes-14-01463-f003:**
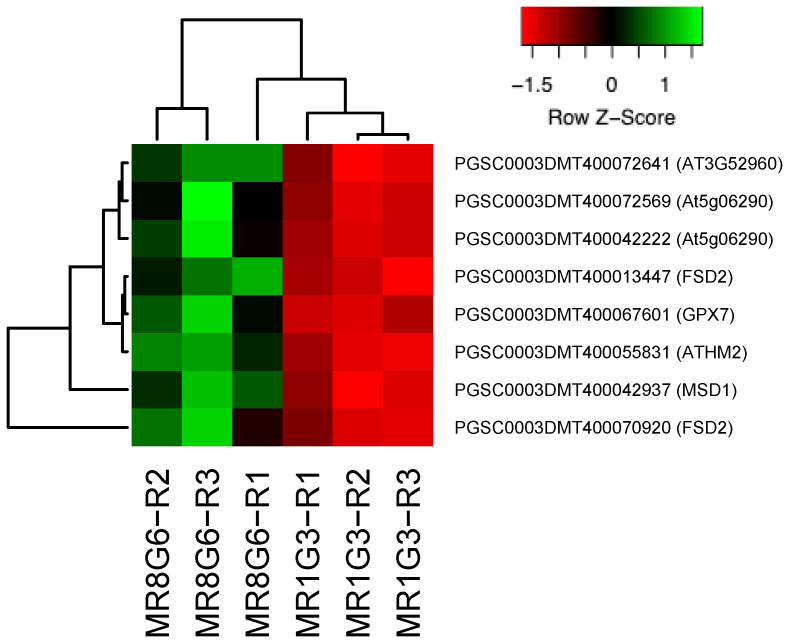
Hierarchical clustering analyses (HCA) and heat map of up-regulated genes involved in redoxins during MTs development under dark conditions; levels of up-regulation are presented in Log2.

**Figure 4 genes-14-01463-f004:**
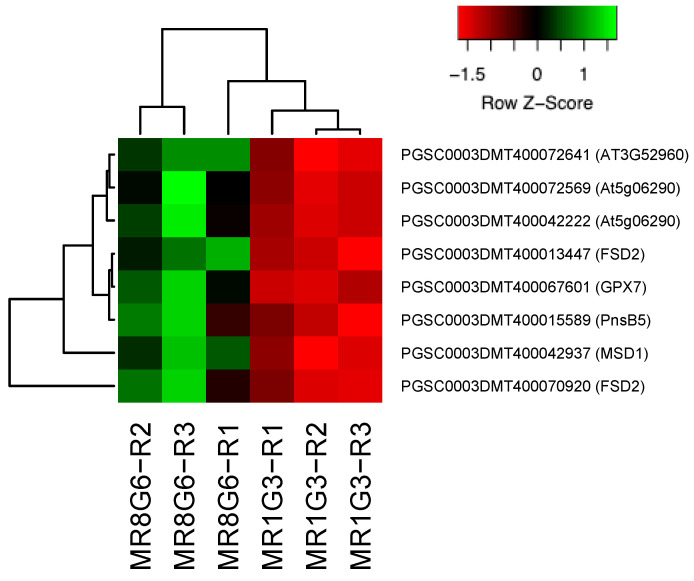
Hierarchical clustering analyses (HCA) and heat map of up-regulated genes involved in oxidative stress during MTs development under dark conditions; levels of up-regulation are presented in Log2.

**Figure 5 genes-14-01463-f005:**
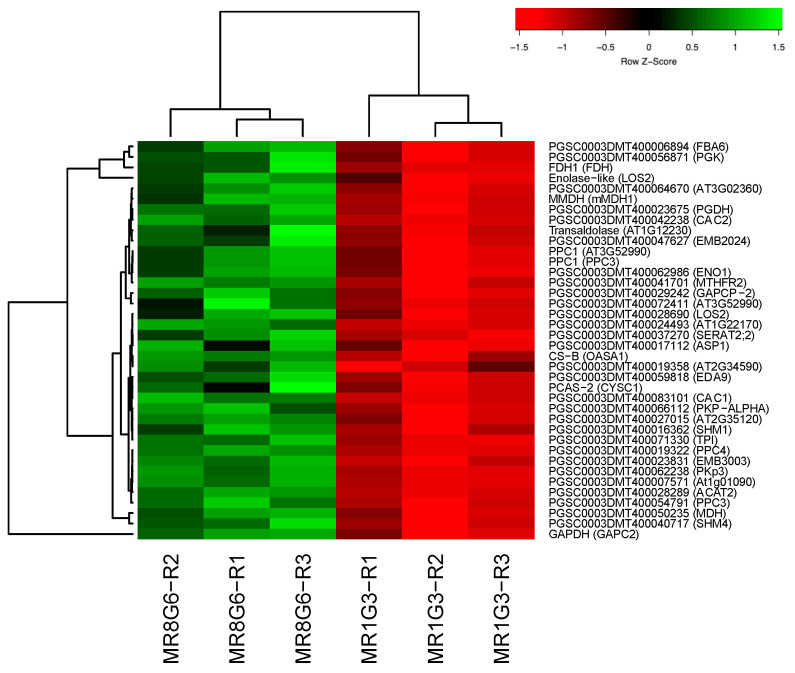
Hierarchical clustering analyses (HCA) and heat map of up-regulated genes involved in carbon metabolism during MTs development under dark conditions; levels of up-regulation are presented in Log2.

**Figure 6 genes-14-01463-f006:**
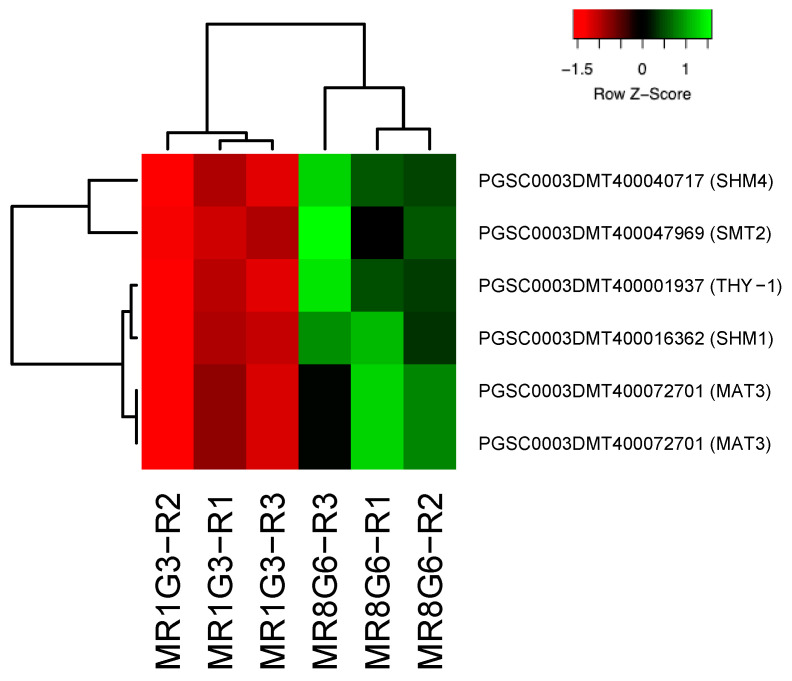
Hierarchical clustering analyses (HCA) and heat map of up-regulated genes involved in one-carbon metabolism during MTs development under dark conditions; levels of up-regulation are presented in Log2.

**Figure 7 genes-14-01463-f007:**
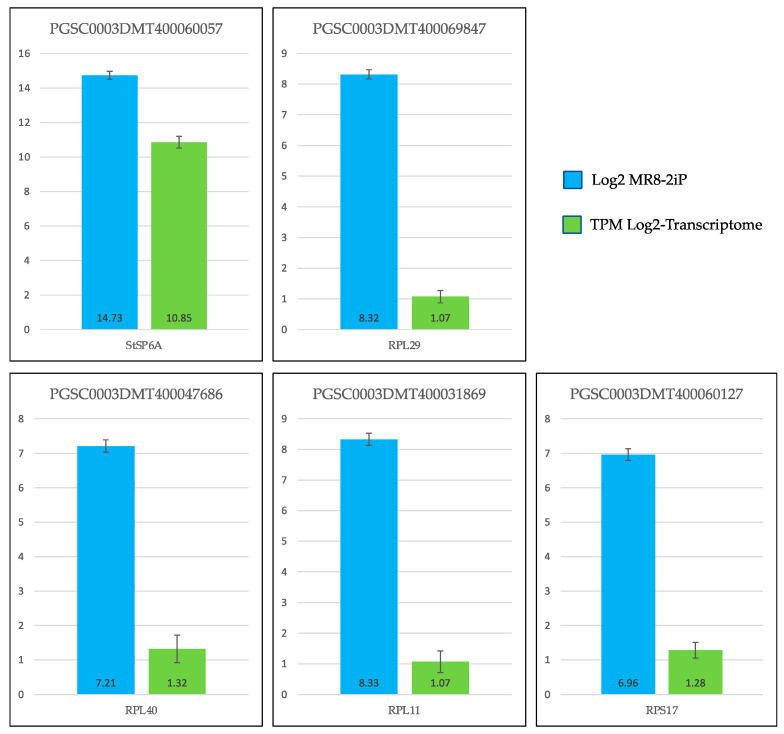
Validation of the transcriptomic-wide analysis by quantitative reverse transcription PCR (qRT-PCR) of 5 DEG up-regulated genes involved in RPs and PEBP of potato *S. tuberosum*. Blue columns correspond to absolute gene expression derived from the genome-wide analysis. The green bars represent the expression of the transcriptome expressing the number of transcripts per million.

**Figure 8 genes-14-01463-f008:**
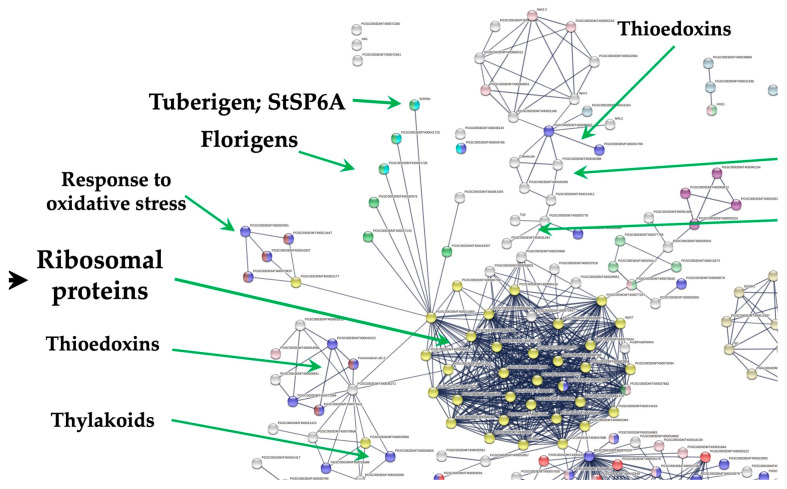
PPI network of RPs interacting with immunophilins, disulfide isomerase, thioredoxins, and oxidative stress proteins during MTs development in potato. RPL29 (PGSC0003DMT400069847) interacts with peptidyl-prolyl cis-trans isomerase *FKBP12*. The PPI network has the highest confidence (0.900).

**Figure 9 genes-14-01463-f009:**
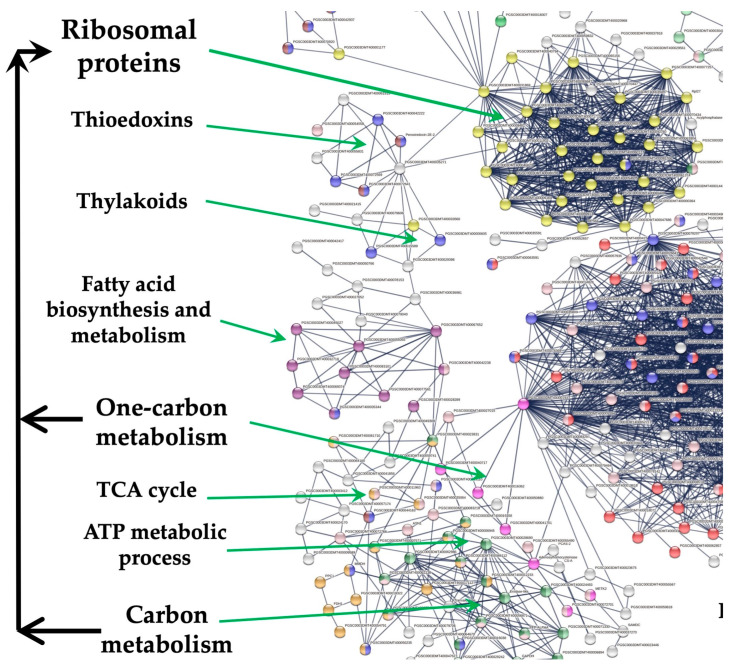
Cluster of genes involved in carbon metabolism, one-carbon metabolism, TCA cycle interacting with the CC cluster and RPs.

**Table 1 genes-14-01463-t001:** Primer design of DEG used to validate the genome-wide analysis of MTs development of potato *S. tuberosum* cv. α Oligonucleotides were designed for qPCR (2^−ΔΔCT^ method analysis) gene expression or transcriptional analysis.

Gen ID	Sequences	ID NCBI
RPL29	F: AATCAGTCGTACAAGGCTCAC	XM_015312492.1
	R: GCATACCTCTGGTTCCTCAAG	
RPL11	F: GAGAAGCTGCCGGATCTTAAT	XM_006343698.2
	R: TCGGAGGGTCAATGTCAATTC	
RPL40	F: AGCCAAGATTCAGGACAAGG	XM_006350430.1
	R: TCAGGACAAGATGCAGAGTTG	
RPL17	F: GTTTCCAATTACCTTGCCGAAG	XM_006345811.2
	R: ACCATTGTTCATCCCGTCTTC	
SP6A	F: TGCAACCTAGGGCTCATATTG	NM_001287968.1
	R: GCCAATGTAGATACTCCCTCAAG	
EFα1	F: TTTGGCCCTACTGGTTTGAC	NM_001288491.1
	R: GCACTGGAGCATATCCGTTT	
SEC3	F: GCTTGCACACGCCATATCAAT	XM_006342542.2
	R: TGGATTTTACCACCTTCCGCA	

**Table 2 genes-14-01463-t002:** Ribosomal proteins with loss of function and chloroplast/ribosome biogenesis factors in plants.

Protein	Function/Process/Organism	Loss-of-Function Phenotype	References
RPs	Ribosomal Proteins		
RPS1	Ribosomal protein. Rice	Seedling lethality	Zhou et al., 2021 [22]
RPS5A	Ribosomal protein. Arabidopsis	Embryo lethality	Weijers et al., 2001 [23]
RPS9	Ribosomal protein. Maize	Embryo lethal	Ma and Dooner, 2004 [24]
RPS13A	Ribosomal protein	Root growth retarded and late flowering	Ito et al., 2000 [25]
RPS16	Ribosomal protein. Arabidopsis	Embryo lethality	Tsugeki et al., 1996 [26]
RPS17	Ribosomal protein. Maize	Seedling lethality	Schultes et al., 2000 [27]
RPS18A	Ribosomal protein. Tobacco	Seedling lethality	Rogalski et al., 2006 [28]
RPS20	Ribosomal protein. Rice	Seedling lethality	Gong et al., 2013 [29]
RPS21	Ribosomal protein. Arabidopsis	Reduced photosynthetic activity	Dong et al., 2020 [30]
RPS27	Ribosomal protein. Arabidopsis	Embryo lethality	Revenkova et al., 1999 [31]
RPL5C	Ribosomal. Arabidopsis	Embryo lethality	Dupouy et al., 2022 [32]
RPL9C, RPL9D	Ribosomal protein. Arabidopsis	Embryo lethality	Devis et al., 2015 [33]
RPL10	Ribosomal protein. Arabidopsis, Maize	Embryo lethality	Falcone et al., 2010 [34]
RPL11	Ribsomal protein. Arabidopsis	Decreased leaf pigmentation, plant growth and photosyntesis	Pesaresi et al., 2001 [35]
RPL12	Ribosomal protein. Rice	Seedling lethality	Zhao et al., 2016 [36]
RPL13	Ribsomal protein. Rice	Embryo lethality	Lee et al., 2019 [37]
RPL15C	Ribosomal protein. Arabidopsis	Embryo lethality	Bobik et al., 2019 [38]
RPL21C	Ribosomal protein. Arabidopsis and Rice	Embryo lethality	Yin et al., 2021, Lin et al., 2015 [39]
RPL23a	Ribosomal protein, Arabidopsis	Abnormal root and leaves, delayed transition to reproductive growth and reduced seed production	Degenhardt and Bonham-Smith, 2008 [40]
RPL24B	Ribosomal protein. Arabidopsis	Defects in Auxin response related to ARF3 and ARF5	Zhou et al., 2010 [41]
RPL28-1	Ribosomal protein. Arabidopsis	Embryo lethality	Romani et al., 2012 [42]
RPL35-1	Ribosomal protein. Maize	Embryo lethality	Magnard et al., 2004 [43]
RPS20, RPL1, RPL4, RPL27 and RPL35	Ribosomal proteins. Arabidopsis	Embryo lethality	Romani et al., 2012 [42]
	Chloroplast/Ribosome biogenesis factors		
EDD1 (GlyRS9)	Glycyl tRNA synthetase. Arabidopsis	Embryo lethality	Uwer et al., 1998 [44]
CFG1, CFG2	Chloroplast development. Arabidopsis	Seedling lethality	Zhu et al., 2020 [45]
DCL-M	Defective chloroplast and leaf- mutable. Tomato	Embryo lethality	Bellaoui et al., 2003 [46]
CPN21	Chaperonin: Tomato, Tobacco	Seed abortion	Hanania et al., 2006 [47]
AtBRX-1-1, AtBRX-1-2	Maturation of the large pre-60S ribosomal subunit	Pointed leaves, delayed growth	Weis et al., 2015 [48]
AtNuc-L1-AtNuc-L2	Ribosome biogenesis. Arabidopsis	Seedling lethality	Durut et al., 2014 [49]
AtTHAL	Nucleolar organization	Embryo lethality	Chen et al., 2016 [50]
AtNMD3	Nuclear export adaptor of 60S pre-ribosome export and maturation	Lethal	Chen et al., 2012 [51]
RID1	DEAH-box RNA helicase, Pre-mRNA splicing	Abnormal shoot and root apical meristem maintenance, leaf and root morphogenesis	Ohtani et al., 2013 [52]
TIC32	Translocon of the inner envelope of chloroplasts	Embryo lethality	Hörmann et al., 2004 [53]
ATS2	Phosphatidic acid as intermediate for chloroplast membrane lipid biosynthesis	Embryo lethality	Yu et al., 2004 [54]
TIC110	Translocon of the inner envelope of chloroplasts	Embryo lethality	Kovacheva et al., 2005 [55]
CHL27	Chlorophyl biosynthesis	Retarded growth and chloroplast developmental defects	Bang et al., 2008 [56]
DG1	Early chloroplast development	Delayed greening phenotype	Chi et al., 2008 [57]
OEP80	Chloroplast outer envelope protein	Embryo lethality	Patel et al., 2008 [58]
EMB5067/AKRP	Embryo development chloroplast protein	Embryo lethality	Garcion et al., 2006 [59]
SPC1	Carotenoid biosynthesis	Embryo lethality	Dong et al., 2007 [60]
PDS3	phytoene desaturase gene,	Embryo lethality	Qin et al., 2007 [61]
EMB1303-1	Chloroplast biogenesis	Embryo lethality	Huang et al., 2009 [62]
EMB1211	Chloroplast biogenesis	Seedling lethality	Liang et al., 2010 [63]
BPG2	Chloroplast protein accumulation induced by Brassinazole	Decreased number of stacked grana thylakoids	Komatsu et al., 2010 [64]
119 Nuclear genes-assoc. w/chloroplast	Embryo defective mutants/associated to chloroplast	Embryo lethality	Bryant et al., 2011 [65]
IRM	Involved in RNA processing	Embryo lethality	Palm et al., 2019 [66]
ZMRH3	The RH3 DEAD Box Helicase	Embryo lethality	Asakura et al., 2012 [67]
HSP90C	Chloroplast biogenesis	Embryo lethality	Inoue et al., 2013 [68]
FTSHI4	Thylakoid membrane-associated protein	Embryo lethality	Lu et al., 2014 [69]
RNAJ	Ribonuclease J (RNase J) required for chloroplast and embryo development	Embryo lethality	Chen et al., 2015 [70]
DER	Chloroplast ribosomal RNA processing	Embryo lethality	Jeon et al., 2014 [71]
Rrp5, Pwp2, Nob1, Enp1 and Noc4	Ribosome biogenesis factors	Embryo lethality	Missbach et al., 2013 [72]
SHREK1	Ribosome biogenesis factor	Embryo lethality	Liu et al., 2022 [73]
NOP2A, NOP2B	tRNA and rRNA methylation profiles	Embryo lethality	Burgess et al., 2015 [74]
RH22	RNA helicase22	Embryo lethality	Chi et al., 2012 [75]
MDN1	The AAA-ATPase MIDASIN 1 functions in ribosome biogenesis	Embryo lethality	Li et al., 2019 [76]

## Data Availability

BioProject: PRJNA898400 https://www.ncbi.nlm.nih.gov/bioproject/PRJNA898400 accessed on 2 June 2023.

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
