# Peer review of "Exploring the Potential Role of Ribosomal Proteins to Enhance Potato Resilience in the Face of Changing Climatic Conditions"

_genes, 2023, doi:10.3390/genes14071463_

Round 1
Reviewer 1 Report
Dear Author,
First of all, this article is not presented in the regular style of scientific articles; all parts of the regular article are not present. Only the introduction and conclusion are present. Material and Methods are not included, while Results and Discussion are presented in the Introduction part. Also, this article is written in such a way that things are going to be done, for example: ... MTs test will be according to Herrera-Isidron et al., 2022 [6], and the protocol of genetic transformation and genome editing will be according to Cabrera-Ponce et al., 1997 [46].
There are minor comments: The scientific names should be written in italics; for example, line 126: Oryza sativa. Also, the author should refer to figs in the text.
Moreover, supplementary materials are not accessible: error 404 - file not found message always comes
English langaue is fine, however there are some typng and syntex mistakes. Authour should revise carefully the text.
Author Response
Answers to Reviewer 1
Dear Author,
First of all, this article is not presented in the regular style of scientific articles; all parts of the regular article are not present. Only the introduction and conclusion are present. Material and Methods are not included, while Results and Discussion are presented in the Introduction part. Also, this article is written in such a way that things are going to be done, for example: ... MTs test will be according to Herrera-Isidron et al., 2022 [6], and the protocol of genetic transformation and genome editing will be according to Cabrera-Ponce et al., 1997 [46].
Answers:
1.- We have made major revisions within the manuscript. It was edited in the regular style.
Within the manuscript we have included:
2.- Materials/Methods and Results about the transcriptomic analysis of microtuberization of potato under darkness.
3.- It was also included The Hierarchical clustering analyses (HCA) and heat map of up-regulated genes that were discussed within the manuscript.
4.- To validate the transcriptomic analysis, we have included the analysis of 5 genes: StSP6A, RPL11, RP29, RPL40 and RPS17. EFa1, SEC3 were used as endogens.
5.- The abstract was modified.
6.- To avoid plagiarism, most of the manuscript was edited.
7.- The Scientific names were written using italics.
8.- Conclusions are not numbered
Reviewer 2 Report
To,
The Chief Editor,
Genes, MDPI,
Manuscript ID: genes-2470833
Subject: Submission of comments on the manuscript in “Genes"
Dear Chief Editor Genes, MDPI,
Thank you very much for the invitation to consider a potential reviewer for the manuscript (ID: genes-2470833). My comments responses are furnished below as per each reviewer’s comments.
Dear Chief Editor,
In the reviewed manuscript, the authors carried out the transcriptomic analysis that revealed that ribosomal proteins have a crucial role during MTs development. The RPLs; RPL11, RPL29 and RPL40 interact with PEBP family members (Phosphatidylethanolamine-binding proteins), poten-tial activators in the MTs development. In potatoes, genome editing is a promising technology to introduce crop breeding traits. Recently, the edition of the genome through the CRISPR/Cas9 system has made it the most efficient and powerful genetic modification method. Based on this, the gene modulation by overexpression and silencing in RPL11, RPL29 and RPL40 will guide us to understand the effect on the microtuberiza-tion process and produce improved potato plants with the capability of growing under adverse environments (biotic and abiotic stresses). By understanding the molecular biology mechanisms that RPs govern, we can improve crops under today's changing climatic conditions. However, in my opinion, the MS needs major revisions. I have some suggestions to improve this manuscript:
- I have read the entire manuscript and my initial comment is that manuscript is poorly written. I have significant concerns about the grammar and vocabulary of the manuscript; therefore, I recommend the authors to use an English proofreading service.
- The structure of the abstract should be improved, as well as the lack of several aspects that should be included in this section. Most of the abstracts contain confusing and uninformative sentences. Please give more precise objectives here (such as in the Abstract). The abstract should highlight the most important results of the parameters and characteristics assayed.
- Keyword must in alphabtical order.
4. The content of the introduction is effective in essence but very poorly presented, significant improvements are needed in presenting the proper background of the work undertaken..
- The figures are quite low resolution and difficult to make out. Higher-resolution versions will be needed for publication. Further, text in figure is not readble. for example, in Figures 2.
- In Material and Methods:- indicate how many replicates assayed in each analysis/parameter. The number of samples or biological and technical replicates should be mentioned for each parameter in the methods.
- Author must validate the transcriptome results by qRT-PCR
- The discussion should be interpreted with the results as well as discussed in relation to the present literature.
- Conclusion section should not be numbering, arrange a paragraph. The author should emphasize this in a better way.
- References: shall have to correct the whole References according to the ”Instructions for the Authors”, e.g. title should not be in italics, the Journal name is in italics, and the author shall have to use the abbreviated name Journals cited the year must be bold, the scientific name must be italics etc. Moreover, duplication of reference numbering. Please check all references carefully.
Best wishes and thank you
To,
The Chief Editor,
Genes, MDPI,
Manuscript ID: genes-2470833
Subject: Submission of comments on the manuscript in “Genes"
Dear Chief Editor Genes, MDPI,
Thank you very much for the invitation to consider a potential reviewer for the manuscript (ID: genes-2470833). My comments responses are furnished below as per each reviewer’s comments.
Dear Chief Editor,
In the reviewed manuscript, the authors carried out the transcriptomic analysis that revealed that ribosomal proteins have a crucial role during MTs development. The RPLs; RPL11, RPL29 and RPL40 interact with PEBP family members (Phosphatidylethanolamine-binding proteins), poten-tial activators in the MTs development. In potatoes, genome editing is a promising technology to introduce crop breeding traits. Recently, the edition of the genome through the CRISPR/Cas9 system has made it the most efficient and powerful genetic modification method. Based on this, the gene modulation by overexpression and silencing in RPL11, RPL29 and RPL40 will guide us to understand the effect on the microtuberiza-tion process and produce improved potato plants with the capability of growing under adverse environments (biotic and abiotic stresses). By understanding the molecular biology mechanisms that RPs govern, we can improve crops under today's changing climatic conditions. However, in my opinion, the MS needs major revisions. I have some suggestions to improve this manuscript:
- I have read the entire manuscript and my initial comment is that manuscript is poorly written. I have significant concerns about the grammar and vocabulary of the manuscript; therefore, I recommend the authors to use an English proofreading service.
- The structure of the abstract should be improved, as well as the lack of several aspects that should be included in this section. Most of the abstracts contain confusing and uninformative sentences. Please give more precise objectives here (such as in the Abstract). The abstract should highlight the most important results of the parameters and characteristics assayed.
- Keyword must in alphabtical order.
4. The content of the introduction is effective in essence but very poorly presented, significant improvements are needed in presenting the proper background of the work undertaken..
- The figures are quite low resolution and difficult to make out. Higher-resolution versions will be needed for publication. Further, text in figure is not readble. for example, in Figures 2.
- In Material and Methods:- indicate how many replicates assayed in each analysis/parameter. The number of samples or biological and technical replicates should be mentioned for each parameter in the methods.
- Author must validate the transcriptome results by qRT-PCR
- The discussion should be interpreted with the results as well as discussed in relation to the present literature.
- Conclusion section should not be numbering, arrange a paragraph. The author should emphasize this in a better way.
- References: shall have to correct the whole References according to the ”Instructions for the Authors”, e.g. title should not be in italics, the Journal name is in italics, and the author shall have to use the abbreviated name Journals cited the year must be bold, the scientific name must be italics etc. Moreover, duplication of reference numbering. Please check all references carefully.
Best wishes and thank you
Author Response
Answers to reviewer 2:
- I have read the entire manuscript and my initial comment is that manuscript is poorly written. I have significant concerns about the grammar and vocabulary of the manuscript; therefore, I recommend the authors to use an English proofreading service.
Dear reviewer 2, we have made major revision within the manuscript.
- The structure of the abstract should be improved, as well as the lack of several aspects that should be included in this section. Most of the abstracts contain confusing and uninformative sentences. Please give more precise objectives here (such as in the Abstract). The abstract should highlight the most important results of the parameters and characteristics assayed.
The abstract was modified
- Keyword must in alphabtical order.
Keywords was listed in alphabetical order
- The content of the introduction is effective in essence but very poorly presented, significant improvements are needed in presenting the proper background of the work undertaken..
Dear reviewer 2, The introduction was improved, I expect that is NOT POORLY for you.
- The figures are quite low resolution and difficult to make out. Higher-resolution versions will be needed for publication. Further, text in figure is not readble. for example, in Figures 2.
Dear reviewer 2, the pictures are perfect for publication, you can apply zoom and be able to read the ID of each gene. You have to understand how to analyzed gene networks.
- In Material and Methods:- indicate how many replicates assayed in each analysis/parameter. The number of samples or biological and technical replicates should be mentioned for each parameter in the methods.
The answers for this were included in Materials/Methods and Results about the transcriptomic analysis of microtuberization of potato under darkness.
It was also included The Hierarchical clustering analyses (HCA) and heat map of up-regulated genes that were discussed within the manuscript.
- Author must validate the transcriptome results by qRT-PCR
To validate the transcriptomic analysis, we have included the analysis of 5 genes: StSP6A, RPL11, RP29, RPL40 and RPS17. EFa1, SEC3 were used as endogens.
- The discussion should be interpreted with the results as well as discussed in relation to the present literature.
The discussion was improved
- Conclusion section should not be numbering, arrange a paragraph. The author should emphasize this in a better way.
It was modified
- References: shall have to correct the whole References according to the ”Instructions for the Authors”, e.g. title should not be in italics, the Journal name is in italics, and the author shall have to use the abbreviated name Journals cited the year must be bold, the scientific name must be italics etc. Moreover, duplication of reference numbering. Please check all references carefully.
References were corrected according to Genes journal
Round 2
Reviewer 1 Report
Dear Author,
The revised version of the MS is greatly improved; however, there are still some minor typing mistakes. You should carefully revise it for the language mistakes.
MS should be revised for typing mistakes.
Reviewer 2 Report
Dear Chief Editor,
Thank you for providing the opportunity to review the revised manuscript. The authors have addressed all comments and incorporated changes suggested by reviewers during the first round of revisions. The revised version of the manuscript is improved as expected. Based on these revisions, now this study is a suitable contribution to the Gene. I recommend the manuscript for publication.
Thank you
With best regards
Dear Chief Editor,
Thank you for providing the opportunity to review the revised manuscript. The authors have addressed all comments and incorporated changes suggested by reviewers during the first round of revisions. The revised version of the manuscript is improved as expected. Based on these revisions, now this study is a suitable contribution to the Gene. I recommend the manuscript for publication.
Thank you
With best regards